# Innate Immune Antagonism of Mosquito-Borne Flaviviruses in Humans and Mosquitoes

**DOI:** 10.3390/v13112116

**Published:** 2021-10-20

**Authors:** Ahmed M. E. Elrefaey, Philippa Hollinghurst, Christine M. Reitmayer, Luke Alphey, Kevin Maringer

**Affiliations:** 1The Pirbright Institute, Pirbright, Woking GU24 0NF, UK; philippa.hollinghurst@pirbright.ac.uk (P.H.); christine.reitmayer@pirbright.ac.uk (C.M.R.); luke.alphey@pirbright.ac.uk (L.A.); 2Department of Microbial Sciences, Faculty of Health and Medical Sciences, University of Surrey, Guildford GU2 7XH, UK

**Keywords:** flavivirus, arbovirus, innate immune signaling, humans, mosquitoes, immune antagonism, type I interferon, RNAi

## Abstract

Mosquito-borne viruses of the *Flavivirus* genus (*Flaviviridae* family) pose an ongoing threat to global public health. For example, dengue, Japanese encephalitis, West Nile, yellow fever, and Zika viruses are transmitted by infected mosquitoes and cause severe and fatal diseases in humans. The means by which mosquito-borne flaviviruses establish persistent infection in mosquitoes and cause disease in humans are complex and depend upon a myriad of virus-host interactions, such as those of the innate immune system, which are the main focus of our review. This review also covers the different strategies utilized by mosquito-borne flaviviruses to antagonize the innate immune response in humans and mosquitoes. Given the lack of antiviral therapeutics for mosquito-borne flaviviruses, improving our understanding of these virus-immune interactions could lead to new antiviral therapies and strategies for developing refractory vectors incapable of transmitting these viruses, and can also provide insights into determinants of viral tropism that influence virus emergence into new species.

## 1. Introduction

Arthropod-borne viruses (arboviruses) of the *Flavivirus* genus (family *Flaviviridae*) are principally transmitted by mosquitoes and ticks, with some flaviviruses having no known vector or being insect-specific viruses with no vertebrate host [1]. Mosquito-borne flaviviruses of major human public health importance, which are the focus of this review, include dengue virus (DENV), Japanese encephalitis virus (JEV), West Nile virus (WNV), yellow fever virus (YFV), and Zika virus (ZIKV). These viruses generally cause a broad spectrum of diseases in humans, including hepatitis, hemorrhage, encephalitis, paralysis, seizures, Guillain-Barre syndrome, and congenital abnormalities [2]. Furthermore, these viruses have been emerging or re-emerging locally or globally over the past several decades and are causing an increasing global public health burden [3,4].

Flaviviruses are enveloped positive-sense single-stranded RNA viruses that encode a single open reading frame flanked by 5′ and 3′ untranslated regions (UTRs) [5]. The genome is translated into a single polyprotein that is cleaved into three structural proteins (capsid, C, premembrane, prM, and envelope, E) and seven non-structural proteins (NS1, NS2A, NS2B, NS3, NS4A, NS4B, and NS5) [5]. Some flaviviruses also encode the 2K peptide, released through proteolytic cleavage events at the NS4A-NS4B junction [6]. The structural proteins constitute the components of the mature assembled virion that mediate viral attachment and entry [5]. The non-structural proteins facilitate genome replication and maturation of the viral genome [7]. The subgenomic flavivirus RNA (sfRNA) is also produced as a result of incomplete degradation of uncapped viral RNA by the host 5′-3′ exoribonuclease Xrn1 [8].

## 2. The Innate Immune Response to Mosquito-Borne Flaviviruses and Its Antagonism in Humans and Mosquitoes

The innate immune response is one of the early defense mechanisms against viral infection in vertebrates [9] and is also a barrier to viral transmission by the vector [10]. Notably, the innate immune response is partially conserved between humans and mosquitoes, particularly for certain signaling cascades. The human innate immune response is mainly comprised of Toll-like receptor (TLR), retinoic acid-inducible gene I (RIG-I)-like receptor, cyclic GMP-AMP synthase (cGAS)-stimulator of interferon genes (STING) signaling, and Janus kinase/signal transducer and activator of transcription (JAK-STAT) pathways. On the other hand, the innate immune response in mosquitoes includes the Toll, immunodeficiency (IMD), JAK-STAT, and RNA interference pathways [11]. Of note, the Toll, IMD, and JAK-STAT pathways were first identified in the model insect species *Drosophila melanogaster* and are highly conserved with those of humans and mosquitoes such as *Aedes aegypti*, which are classified in the order *Diptera* (true flies) along with *D. melanogaster* [10,12]. Mosquito-borne flaviviruses evade these pathways in mosquitoes to establish persistent infection and allow their dissemination from the initial site of infection in the midgut into the mosquito’s open circulatory system (hemocoel) and the salivary glands [13]. Moreover, the pathogenesis of some flaviviruses in vertebrates is driven at least in part by a dysregulation of the immune system, including early innate immune responses. The molecular mechanisms of viral antagonism of these pathways are well studied in humans; however, comparable studies in mosquitoes are relatively less advanced.

This review highlights similarities and differences in the major innate immune responses to mosquito-borne flaviviruses in the vertebrate host and invertebrate vector. We also focus on the known molecular mechanisms by which mosquito-borne flaviviruses antagonize the antiviral innate immune responses that are conserved between their mosquito vectors and the human host. Besides encoding immune antagonists, several other strategies are utilized by mosquito-borne flaviviruses to evade the innate immune response in humans and/or mosquitoes [14,15,16,17,18,19,20]. For instance, the genetic composition of humans and dual-host flaviviruses are underrepresented in CpG dinucleotides compared to the largely unbiased frequencies of CpG in the mosquito genome and insect-specific flaviviruses [21,22]. Recent findings have revealed that human zinc-finger antiviral protein (ZAP) is a restriction factor preventing the replication of insect-specific viruses in human cells as it binds CpG in the viral genome [23]. While this and other mechanisms of evading immune detection are interesting, we here focus on the antagonism of immune responses downstream of immune detection.

### 2.1. Nuclear Factor Kappa-Light-Chain-Enhancer of Activated B Cells (NF-κB) and Interferon Regulatory Factor (IRF)3/7-Mediated Immune Responses and Their Antagonism by Mosquito-Borne Flaviviruses in Humans and Mosquitoes

Following the detection of mosquito-borne flaviviruses in human cells, the innate immune signaling response is activated, leading eventually to the expression of proinflammatory cytokines and type I IFN (Figure 1). In mosquitoes, the expression of antimicrobial peptides (AMPs) and other gene products is induced following viral infection through different signaling cascades (Figure 2).

#### 2.1.1. NF-κB and IRF3/7 Signaling Antagonism by Mosquito-Borne Flaviviruses in Humans

Mosquito-borne flaviviruses block the activation of different signaling molecules including those of TLR signaling in humans (Figure 1). For example, TLR3 signaling is inhibited by NS1 from a human isolate of the WNV Texas strain [34,35,36]; however, other studies failed to observe a similar finding following either infection with the virulent WNV New York strain (WNV_NY99_) or the ectopic expression of WNV_NY99_ NS1 [37,38]. Moreover, NS1 of the WNV Uganda strain and other flaviviruses, such as DENV-2 or YFV, are also unable to inhibit TLR3 signaling [38]. This discrepancy might be related to differences in the experimental approaches and viral strains.

*N*-linked glycosylation of mosquito-borne flaviviruses has a great impact on viral virulence and immune evasion as protein glycosylation is divergent between the mammalian and insect cells, particularly in terms of *N*-linked glycans [39,40]. For example, the glycosylation pattern of WNV virulent strain (CT-2741) and recombinant WNV E protein derived from mosquito but not mammalian cells prevents the production of dsRNA-induced cytokines in human cells [41]. The dipteran-glycosylated form of WNV E inhibits TLR3 signaling at the level of the downstream kinase RIP1, preventing both RIP1 ubiquitination and subsequently NF-κB activation in human cells [41].

When viral RNA is sensed by RIG-I in human cells, RIG-I is activated through a polyubiquitination step, an activity mediated by the E3 ligase tripartite motif 25 (TRIM25) [42]. The sfRNAs of ZIKV and DENV-2 have been shown to bind and inhibit the deubiquitylation of TRIM25, which, in turn, inhibits RIG-I activation [43]. The methyltransferase domain of ZIKV NS5 also binds RIG-I and represses its signaling [44]. Moreover, ZIKV sfRNA suppresses MDA5-mediated signaling; however, the exact mechanism is yet to be identified [45]. Interestingly, the wing domain of WNV NS1 resembles the helicase domain of RIG-I and MDA5 [46], and NS1 has been found to physically interact and block the expression of MDA5 and RIG-I [47].

In addition to PRR antagonism, mosquito-borne flaviviruses can antagonize the downstream adaptor and signaling molecules in human cells (Figure 1). The 14-3-3 protein family regulates RLR-mediated antiviral signaling by promoting RIG-I and MDA5 translocation from the cytosol to the mitochondrial membrane through 14-3-3ε and 14-3-3η, respectively [48]. In a proteolysis-independent manner, NS3 of ZIKV and DENV binds 14-3-3ε, which, in turn, impedes RIG-I translocation to the mitochondrion-localized MAVS [49,50]. The inhibitory activity of NS3 is mapped to its RxEP motif for DENV and RLDP motif for ZIKV, both of which mimic the binding motif of 14-3-3 protein and hence compete with RIG-I and MDA5 for 14-3-3 binding [50]. This NS3 motif is partially conserved across several mosquito-borne flaviviruses, including DENV, WNV, and ZIKV; however, YFV and JEV lack this motif [49]. In addition, two mitofusin proteins (MFN1 and MFN2) are required for RIG-I signaling and maintenance of the mitochondrial membrane potential respectively in human cells, and DENV NS2B/3 has been found to cleave both proteins, suppressing RIG-I signaling [51]. Furthermore, ZIKV and DENV NS4A binds MAVS and disrupts its interaction with RIG-I and MDA5 [52,53], and ZIKV NS2B/3 directs MAVS for proteasomal degradation [54].

Mosquito-borne flaviviruses block the downstream signaling cascades at different levels (Figure 1). For example, DENV NS2B/3, in both active and inactive forms, interacts with and masks the kinase domain of IKKε, thereby inhibiting IRF3 phosphorylation and nuclear translocation [55]. Additionally, NS2A and NS4B of DENV and NS4B of WNV inhibit IFN-β induction by blocking TBK1 phosphorylation [56]. NS4A of DENV-1 but not DENV-2 or -4 inhibits TBK1-directed IFN-β induction [56]. In the case of ZIKV, NS1, NS2A, and NS4B have been shown to physically interact with TBK1, blocking its phosphorylation and/or oligomerization [57]. NS5 of the African ZIKV strain MR766 blocks TBK1 interaction and thereby IRF3 activation; however, NS5 of the Cambodian ZIKV strain (FSS13025) did not show a similar inhibitory effect [52,58]. However, this difference in TBK1 antagonism may be an artifact of the repeated passage of ZIKV MR766 strain in suckling mouse which may lead to relative divergence at the amino acid level of NS5 between these strains [59]. NS5 of ZIKV Asian strains have been shown to block the phosphorylation of both IKKε and IRF3 [44,60]. JEV NS5 also blocks IRF3 and NF-κB activation by inhibiting their nuclear translocation through competitive binding to importins α3 and α4, which are nuclear transport proteins required for nuclear import of IRF3 and NF-κB [61,62]. Moreover, the presence of JEV sfRNA in virus-infected cells results in decreased IRF-3 phosphorylation [63].

NS2B of DENV-2 and DENV-4 is able to antagonize the cGAS-STING pathway by promoting cGAS degradation in an autophagy-lysosome-dependent manner [64]. Moreover, inhibition of STING is widely observed for many mosquito-borne flaviviruses, including DENV, JEV, WNV, and ZIKV in human cells [54,65]. Of note, NS2B/3 of these viruses has been shown to cleave human STING, but not its murine homolog, implying the relevance of the innate immune system in shaping the vertebrate host tropism of mosquito-borne flaviviruses [66,67]. For instance, DENV fails to replicate efficiently in some nonhuman primates probably due to the inability of its NS2B/3 to inactivate STING [68]. Furthermore, YFV NS4B has been shown to bind and inhibit STING, an activity that has been hypothesized to be related to a shared homology between the N-terminus of STING and a number of flavivirus NS4Bs [69].

#### 2.1.2. NF-κB Signaling Antagonism by Mosquito-Borne Flaviviruses in Mosquitoes

Few studies have focused on the identification of innate immune antagonists of mosquito-borne flaviviruses in mosquitoes (Figure 2). DENV-2 infection has been shown to generally suppress immune pathways in mosquito cells and in vivo [70,71,72]. For instance, the increased production of DENV-2 sfRNA could inhibit the expression of the NF-κB family transcription factor Rel1A, particularly in mosquito salivary glands [71]. In contrast, ZIKV and WNV sfRNAs do not inhibit the Toll pathway [73]. Additionally, DENV-2 NS4B inhibits the expression of antimicrobial peptides in insect cells through the activation of the unfolded protein response (UPR), a pathway that is involved in the crosstalk between immune pathways [74].

**Figure 2 viruses-13-02116-f002:**
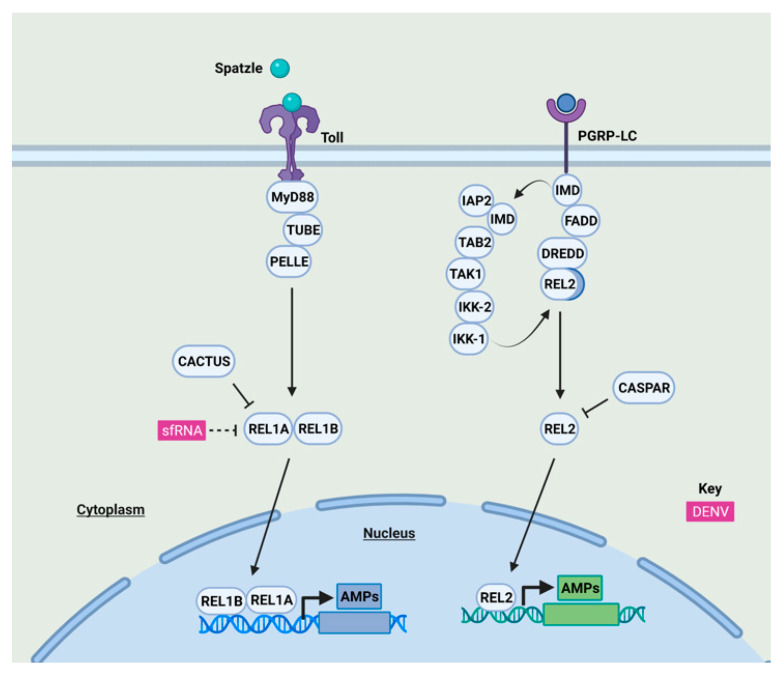
NF-κB signaling in response to mosquito-borne flavivirus infection and its antagonism in mosquitoes. The Toll pathway is activated when the ligand spätzle binds Toll receptors, which recruits MyD88, Tube, and Pelle, resulting in proteasomal degradation of the negative regulator Cactus; therefore, leading to the nuclear translocation of Rel1A and its co-activator Rel1B [75,76]. The IMD pathway is activated through peptidoglycan receptors (PGRP-LC), which promotes the recruitment of adaptor molecules such as IMD and Fas-associated death domain (FADD) proteins which, in turn, leads to the activation of the caspase Dredd [75]. Then, the NF-κB family transcription factor Rel2 is cleaved by Dredd [75]. IMD is cleaved by caspases which allow IMD to bind the inhibitor of apoptosis (IAP2), activating the IKK complex, allowing the nuclear translocation of the N-terminal domain of Rel2 [75]. Caspar is the negative regulator of Rel2 [76]. Rel1A and Rel2, homologs of *Drosophila* Dorsal and Relish respectively [77], induce the expression of antimicrobial peptides and other immune-regulated genes [11,75]. How the Toll and IMD pathways detect viral infection is not yet known. Although a cGAS-like receptor (cGLR)-STING signaling pathway has recently been characterized in *D. melanogaster* [78,79,80,81], this pathway appears to be absent from mosquitoes [82]. sfRNA of DENV, in purple, is able to antagonize the Toll pathway. See main text for details.

### 2.2. JAK-STAT Signaling and Its Antagonism by Mosquito-Borne Flaviviruses in Humans and Mosquitoes

IFNs are a family of cytokines that play an important role in controlling viral infection in humans, and they are classified into type I, II, and III IFNs [83]. Although Type I and III IFNs showed antiviral effects against mosquito-borne flaviviruses [2], our main focus here is on the antiviral role of type I IFN as it is well characterized and studied in terms of signaling and immune evasion of mosquito-borne flaviviruses (Figure 3A). While the JAK-STAT signaling pathway is conserved in insects, no IFN orthologs have been identified in mosquitoes [84]. However, both human and mosquito species possess a similar JAK-STAT signaling cascade (Figure 3B).

#### 2.2.1. JAK-STAT Signaling Antagonism by Mosquito-Borne Flaviviruses in Humans

Once a successful viral replication cycle is achieved in humans, IFN treatment does not restrict the replication of mosquito-borne flaviviruses such as DENV, WNV, and ZIKV [93,94,95], which implies that these viruses encode antagonists that can circumvent the antiviral effect of IFN (Figure 3A). WNV inhibits the JAK-STAT pathway in humans at an early phase. WNV NS5 down-regulates the surface expression of IFNAR1 by binding prolidase, an enzyme required for IFNAR1 trafficking and surface expression; therefore, prolidase deficiency is associated with reduced IFNAR1 maturation and IFN signaling [96]. Mosquito-borne flaviviruses also interfere with different signaling molecules of the JAK-STAT pathway in humans. For example, DENV-2 NS2B/3 and JEV NS5 suppress Tyk2 phosphorylation [97,98]. Furthermore, ZIKV induces proteasomal degradation of JAK, an activity mediated via the helicase domain of NS3 [57].

Following the activation of JAK-STAT signaling in human cells, mosquito-borne flaviviruses block STAT2 function through various conserved and virus-specific mechanisms (Figure 3A). STAT2 degradation is a conserved strategy mediated by the NS5 of diverse mosquito-borne flaviviruses but via different mechanisms. For example, DENV NS5 binds STAT2 and facilitates its degradation via ubiquitin-dependent proteolysis [99,100,101], which is mediated by the E3 ubiquitin-protein ligase UBR4, a member of the N-recognin family that targets proteins for degradation [102]. UBR4 has been shown to bind DENV NS5 only following its processing from a polyprotein precursor but not when NS5 is expressed as a stand-alone protein [103]. On the other hand, NS5 of ZIKV and WNV physically interact with and sequester STAT2 via a UBR4-independent manner [95,104,105,106]. Although the NS5s of DENV, ZIKV, and YFV are all able to sequester STAT2, for YFV NS5 this activity is dependent on interferon stimulation and NS5 ubiquitination by TRIM23 [107], but for DENV and ZIKV NS5 this activity is IFN-independent [100]. In addition, the NS5 of JEV and YFV down-regulate STAT2 phosphorylation (99, 109), in contrast to the NS5 of JEV and WNV, which prevent STAT1 phosphorylation [98,105]. A recent study by Wang and colleagues also showed that the methyltransferase and RNA-dependent RNA polymerase domains of DENV and ZIKV NS5 form an interdomain cleft where the coiled-coil domain of STAT2 is anchored, impeding STAT2’s interaction with IRF9 [108]. Notably, DENV, YFV, and ZIKV replicate efficiently in human cells but not mouse cells as their NS5 cannot target mouse STAT2 for degradation, suggesting that STAT2 antagonism contributes to the host specificity of mosquito-borne flaviviruses [88,106]. In line with this, the introduction of human STAT2 into the mouse STAT2 locus produced an immunocompetent mouse model for ZIKV infection [109].

In addition to NS5, DENV-2 encodes additional STAT antagonists, as the collective activity of NS4B, NS4A and NS2A has been found to abrogate STAT1 phosphorylation in response to IFN-β stimulation [98]. Interestingly, two human DENV-1 isolates with V116A or V116M sequence differences in their NS4B proteins (relative to wild-type DENV-1 NS4B) showed enhanced viral growth and lower levels of IFN-β induction relative to wild-type DENV-1 [110]. Moreover, JEV NS4A lacking the C-terminal 2K domain can inhibit the phosphorylation of STAT1/2 heterodimers but not Tyk2 in IFN-treated cells [110].

#### 2.2.2. JAK-STAT Signaling Antagonism by Mosquito-Borne Flaviviruses in Mosquitoes

Mosquito-borne flaviviruses such as WNV and JEV have been shown to antagonize STAT proteins in mosquitoes (Figure 3B), although the mechanisms and implications remain relatively uncharacterized. WNV NS5 and NS1 facilitate the proteasomal degradation of *Culex quinquefasciatus* STAT by upregulating the E3 ubiquitin ligase cullin 4 [111]. In addition, JEV infection suppresses STAT phosphorylation in mosquito (*Aedes albopictus*) cells [92].

### 2.3. RNAi Pathways and Their Antagonism by Mosquito-Borne Flavivirus in Humans and Mosquitoes

RNAi is a highly conserved evolutionary pathway across species from different kingdoms (fungi, plants, and animals) in which small interfering RNAs (siRNAs), microRNAs (miRNAs), and piwi-interacting RNAs (piRNAs) regulate endogenous gene expression and/or provide antiviral immunity [112,113,114]. While the type I IFN response is one of the important components of the antiviral innate immunity in human cells, RNAi is the most well-studied antiviral innate immune pathway in insects. RNAi also possesses an antiviral role in human neural progenitors and pluripotent stem cells [115,116] as antiviral Dicer (aviD), an isoform of Dicer has been shown to protect stem cells from RNA viral infections [117]; however, RNAi becomes inactive in differentiated human cells in which the IFN response is active, which highlights intrinsic differences in innate immunity between pluripotent and differentiated cells [118,119]. Notably, the IFN pathway suppresses RNAi in differentiated cells through the expression of certain ISGs [120] and via laboratory of genetics and physiology 2 (LGP2)-dependent inhibition of Dicer [121]. The similarities and differences in the antiviral RNAi response between humans and mosquitoes are reviewed in more detail in [119]. Here we focus mainly on the well-characterized exogenous siRNA (exo-siRNA) pathway in mosquitoes, and its antagonism by mosquito-borne flaviviruses (Figure 4).

#### 2.3.1. The Exo-siRNA Pathway Antagonism by Mosquito-Borne Flaviviruses in Humans

It is under debate whether mosquito-borne flaviviruses encode viral suppressors of RNAi (VSRs) that function in human cells. However, mosquito-borne flaviviruses such as ZIKV can infect the placenta and fetal brain by targeting neural progenitor cells, impairing their differentiation and proliferation [122,123]. ZIKV has been shown to induce RNAi response in human neural progenitors as it generates vsiRNAs that are lost upon cell differentiation into neural cells [116]. A recent study showed that ZIKV capsid binds Dicer and blocks its activity in neural progenitor cells (Figure 4A), which leads to a global suppression of miRNA production [124]. Therefore, further studies are needed to unravel whether Dicer blockade may block the putatively antiviral siRNA pathway in humans as both pathways share the same Dicer gene.

On the other hand, some studies have reported RNAi pathway suppression by mosquito-borne flaviviruses in differentiated human cells (Figure 4A). For example, NS2A of DENV-2, JEV, WNV, and ZIKV has been shown to possess bona fide VSR activity irrespective of type I IFN production in human somatic cells and mice, as DENV-2 mutants in which NS2A was unable to antagonize RNAi showed attenuated viral replication and infection, which was successfully rescued by silencing Dicer [125]. Besides, DENV-2 NS4B interferes with the cleavage of siRNAs by Dicer in an in vitro assay, although the exact mechanism remains unclear [126]. Furthermore, DENV-1 and WNV sfRNAs suppress the RNAi pathway in human somatic cells as they competitively interfere with the Dicer-mediated processing of dsRNAs, suggesting that the sfRNA acts as a Dicer decoy substrate [127,128].

#### 2.3.2. The Exo-siRNA Pathway Antagonism by Mosquito-Borne Flaviviruses in Mosquitoes

Mosquito-borne flaviviruses encode VSRs that antagonize vsiRNAs production in mosquitoes (Figure 4B) [129,130]. For example, the C protein of multiple mosquito-borne flaviviruses, including DENV, WNV, YFV, and ZIKV showed a conserved ability to inhibit dsRNA cleavage by Dicer-2 in *A. aegypti* when expressed recombinantly by Sindbis virus [131]. Similar to their role in mammalian cells, NS2A of DENV and JEV suppress the RNAi machinery in mosquito cells and protect the terminal regions of the viral dsRNA from Dicer-2 cleavage in mosquito (*A. aegypti*) cells [125]. The sfRNA of DENV-1 and WNV_KUNV_ blocks the RNAi pathway by interacting with Ago2 and Dicer-2, which in turn, inhibits dsRNA cleavage in the RNAi-competent mosquito (*A. albopictus* and *D. melanogaster*) cells and *C. quinquefascitus* mosquitoes [127,128]. However, another study showed that the replication of wild-type WNV (a southeastern Europe isolate) and its sfRNA-deficient virus showed similar levels of replication in infected *Culex pipiens* mosquitoes [132]. ZIKV sfRNA has been demonstrated as a requirement for virus transmission by mosquitoes by overcoming the mosquito midgut and salivary glands barriers [133]. ZIKV sfRNA has been shown to suppress siRNA production in vivo [73]. However, while wild-type ZIKV exhibits enhanced replication in *A. aegypti*-derived Aag2 cells lacking Dicer-2 (but not cells lacking Ago2) [134], no differences in replication were observed for sfRNA-deficient ZIKV compared to wild-type virus in the RNAi-deficient *A. albopictus*-derived C6/36 cell line [133].

## 3. Conclusions

The innate immune response against mosquito-borne flaviviruses poses an important barrier that limits the replication and spread of viral infection, as well as influencing the host tropism of flaviviruses. In contrast to the human innate immune response, which is normally able to eliminate and clear viral infection, mosquitoes only limit viral dissemination and transmission without clearing the viral infection. Thus, revealing the intricate interplay between flavivirus-encoded immune antagonists and antiviral host effector molecules is crucially important to improve our understanding of viral pathogenicity and disease severity in humans and viral persistence and fitness costs in mosquitoes. While the molecular immune antagonism mechanisms of mosquito-borne flaviviruses are well studied in humans, these studies are less well developed in mosquitoes, particularly for the innate immune signaling cascades. An interesting question is to investigate whether the flavivirus-encoded immune antagonists of mosquito-borne flaviviruses display similar antagonistic activity against the evolutionarily conserved signaling pathways in human and mosquito cells. Another key question is to determine whether these antagonistic activities are mediated by distinct protein domains from the separate roles of these viral proteins in viral replication and maturation. Particularly in the vector, flaviviral antagonists of the immune system could prove to be useful tools for elucidating the molecular events contributing to innate immune function, which remain incompletely characterized in mosquitoes compared to the distantly related model organism *D. melanogaster*. Furthermore, insect NF-κB and Jak-STAT responses are poorly understood in terms of how they may detect and control viral infection relative to the RNAi pathway’s antiviral functions, and in this respect studying flaviviral immune antagonists may also help provide much needed new insights into basic immune functions.

Interestingly, while some flavivirus-encoded immune antagonists are conserved in their ability to antagonize innate immunity across different flaviviruses, other flavivirus-encoded immune antagonists function differently in different viral species. Elucidating how these molecular differences are driven by the evolutionary arms race between virus and host might provide insight into viral emergence and could be exploited for rational vaccine design. On a practical level, it may be possible to develop improved small animal models for flaviviral diseases by modifying their innate immune system to be better antagonized by flaviviruses (thus allowing enhanced viral replication in an immunocompetent background). Such models would be transformative tools to facilitate the development of much-needed antiviral drugs and vaccines. Meanwhile, modifying the vector’s immune system is a potential strategy for developing refractory mosquitoes that cannot transmit flaviviruses. Overall, future developments in this area have the potential to help reduce the increasing global public health burden of flaviviral diseases while improving the preparedness for the potential future emergence of new flaviviral pathogens.

## Figures and Tables

**Figure 1 viruses-13-02116-f001:**
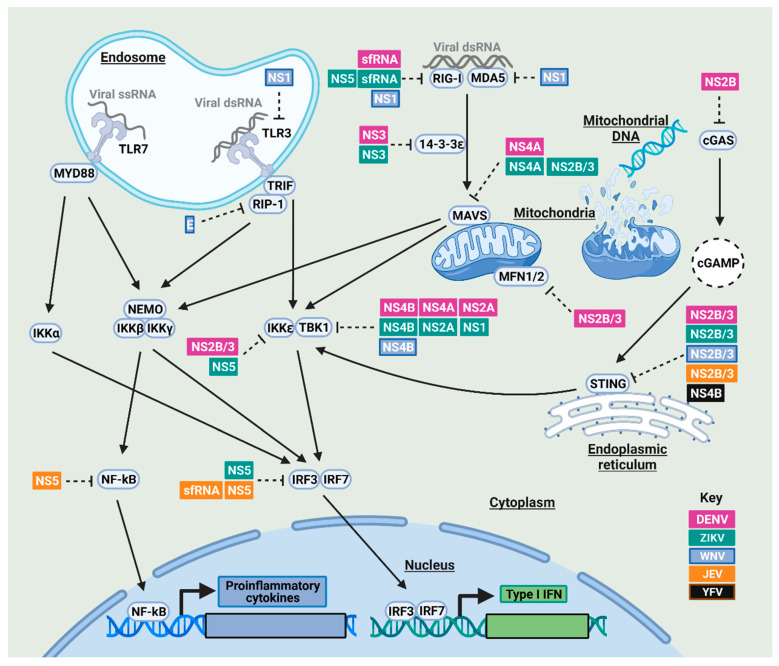
NF-κB and IRF signaling in response to mosquito-borne flavivirus infection and their antagonism in humans. Following viral infection, human cells recognize pathogen-associated molecular patterns (PAMPs) such as viral RNA through pattern recognition receptors (PRRs) that include membrane-bound TLRs such as TLR-3 and -7 and cytoplasmic members of the RIG-I-like receptor (RLR) family, including RIG-I and melanoma-differentiation-associated protein 5 (MDA5) [24,25]. In TLR signaling, TLR-3 and -7 detect double-stranded and single-stranded viral RNAs, respectively. Activated TLR-3 and -7 signal through the adaptor proteins Toll/interleukin receptor (TIR)-domain-containing adapter-including interferon (IFN)-β (TRIF) and myeloid differentiation primary response 88 (MyD88), respectively. On the other hand, RLR signaling is activated once double-stranded RNAs (dsRNAs) are sensed by RIG-I and MDA5, which, in turn, allows these receptors to bind the mitochondrial antiviral signaling (MAVS) protein through 14-3-3ε [26]. In addition, the cytosolic sensor cGAS is able to detect mosquito-borne flaviviruses as they disrupt mitochondrial morphodynamics, leading to the release of mitochondrial DNA that acts as a ligand for the activation of the cGAS-STING pathway [27,28]. cGAS induces the production of cGAMP which then binds to STING. Taken together, signaling through TRIF, MyD88, MAVS, and STING leads to the activation of different signaling molecules, including receptor-interacting protein (RIP)-1, NF-κB essential modular (NEMO), downstream kinases such as Tumor necrosis factor receptor-associated factor (TRAF) family member-associated NF-κB activator (TANK)-binding kinase 1 (TBK1), I-kappa-B-kinase ε (IKKε), and the IKK-αβγ complex, which subsequently activate transcription factors that include NF-κB, IRF3, and/or IRF7 [29,30,31]. Consequently, NF-κB, IRF3, and/or IRF7 are translocated to the nucleus where they elicit the production and release of proinflammatory cytokines and type I IFN, respectively [32,33]. However, virus-encoded antagonists of mosquito-borne flaviviruses (DENV in purple, ZIKV in green, WNV in light blue, JEV in orange, YFV in black) are able to block these pathways. See main text for details.

**Figure 3 viruses-13-02116-f003:**
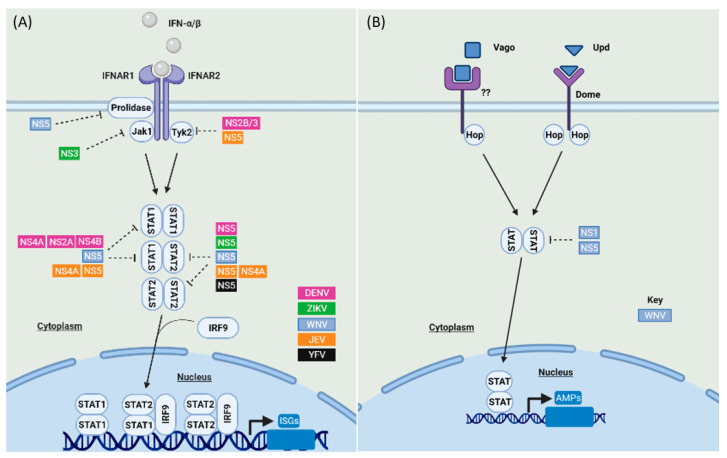
JAK-STAT signaling in response to mosquito-borne flavivirus infection and its antagonism in humans (**A**) and mosquitoes (**B**). (**A**) Following their production and release from human cells, type I IFNs (IFN-α and -β) bind to their cognate IFN-α receptor subunits 1 or 2 (IFNAR1 and IFNAR2) [85]. IFNAR activation results in phosphorylation of JAK1 and tyrosine kinase 2 (Tyk2) and the subsequent activation and homo- or heterodimerization of STAT1 or STAT2 [86]. STAT proteins then associate with IRF9 forming the interferon-stimulated genes factor 3 (ISGF3) complex which is translocated to the nucleus, promoting the transcription of IFN-stimulated genes (ISGs) [87,88]. (**B**) The JAK-STAT pathway in mosquitoes is activated upon the binding of the unpaired ligand (Upd) to the transmembrane receptor Domeless (Dome), an ortholog of the mammalian type I cytokine receptor during development [10,89]. The JAK-STAT pathway is also activated by the binding of Vago, an insect cytokine-like factor, to an unknown receptor which may suggest that Vago acts in a manner similar to IFNs in human cells [90]. Notably, Vago is induced via the RNAi pathway dsRNA sensor Dicer-2, indicating a cross-talk between these pathways in mosquitoes [91]. The binding of Upd and Vago to their cognate receptors leads to the phosphorylation of Hop, a homolog of the mammalian kinase JAK [10], which then results in phosphorylation of mosquito STAT proteins, which show a high degree of similarity to mammalian STAT5 [92]. STAT proteins are then translocated to the nucleus and consequently induce the transcription of JAK-STAT-regulated genes [76]. However, virus-encoded antagonists of mosquito-borne flaviviruses (DENV in purple, ZIKV in green, WNV in light blue, JEV in orange, YFV in black) are able to block these pathways in humans and mosquitoes. See main text for details.

**Figure 4 viruses-13-02116-f004:**
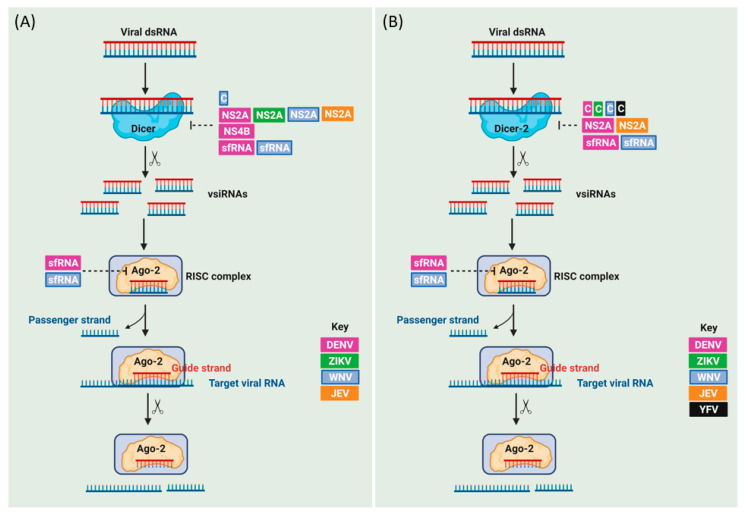
The exo-siRNA pathway in response to mosquito-borne flavivirus infection and its antagonism in humans (**A**) and mosquitoes (**B**). dsRNAs generated during viral replication in mosquitoes are recognized by the PRR Dicer-2 which cleaves the dsRNAs into 21-nucleotide virus-derived siRNAs (vsiRNAs) [122]. The vsiRNA duplex is then incorporated into Argonaute proteins (Ago2) leading to the formation of the RNA-induced silencing complex (RISC) [122]. The passenger strand of the vsiRNA duplex is eliminated from the activated RISC, while the guide strand is retained, allowing recognition of matching viral RNA sequences based on Watson-Crick base pairing, and then complementary RNAs are degraded by the endonuclease activity of Ago-2 [122]. Note that miRNAs are processed by Dicer-1 in insects, separating the miRNA and antiviral exo-siRNA pathways in insects through duplication of the Dicer gene [123]. In contrast, human RNAi is mediated by a single Dicer gene that recognizes dsRNA structures from siRNA and miRNA pathways [124]. Virus-encoded antagonists of mosquito-borne flaviviruses (DENV in purple, ZIKV in green, WNV in light blue, JEV in orange, YFV in black) are able to block these pathways in humans and mosquitoes. See main text for details.

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
