# Peer review of "Innate Immune Antagonism of Mosquito-Borne Flaviviruses in Humans and Mosquitoes"

_viruses, 2021, doi:10.3390/v13112116_

Round 1
Reviewer 1 Report
The authors present a state-of-the-art summary of documented antagonisms by proteins of mosquito-borne flaviviruses in humans and mosquitoes. These antagonisms favour viral replication and the spread of infection. Deeper research into these interactions would aid in finding new therapies and strategies to decrease the public health burden of mosquito-born flaviviral diseases.
The juxtaposition of what is known about the antagonism of innate immunity in humans vs. in mosquitoes highlighted clearly that this topic is shockingly understudied in mosquitoes. Hence, this manuscript is understandably more human-focused.
Comments:
In-text citations should be reviewed. There seems to be an error with the reference manager software has occurred during the formatting of the manuscript.
A table listing all acronyms would be very helpful to the reader.
Line 86. Underscore instead of space.
Figure 1 caption. Several acronyms appear for the first time but have not been defined: IRF, TIR, TANK.
Figures 1 and 2 caption titles could be more specific to what is being depicted, i.e., not all innate immune detection, but just the NF-KB signaling antagonism.
Figure 2 might be better placed after Section 2.1.2.
Figure 2 depicting the mosquito NF-KB immune pathways is incomplete. The mosquito Toll pathway should include the ligand Spaetzle and NF-KB inhibitor Cactus. The IMD pathway transmembrane receptor is PGRP-LC and has an NF-KB inhibitor Caspar. That Rel is short for Relish should be mentioned.
Line 145. The acronym MTase should be defined.
Line 194-195. “Furthermore, the N-terminal sequence of STING shares homology with NS4B of YFV and DENV, and YFV NS4B has been shown to inhibit STING activity”. What does this shared homology imply? Is the ability of YFV NS4B to inhibit STING related to this shared homology?
Line 251- 283. In general, the flow of information in this section is not the most intuitive and jumps around between STAT1 and STAT2, and across viruses, e.g., “STAT2-mediated sequestration by NS5 of DENV and ZIKV occurs regardless of IFN stimulation” and “the collective activity of DENV-2 NS4B, and to a lesser extent, NS4A and NS2A has been found to abrogate STAT1 phosphorylation in response to IFN-β” seem like they should go together. Admittingly, these are complex interactions to describe but it would read better if the authors could find a more streamlined way to present it.
Line 255. The acronym UBR4 should be defined.
Line 263-268. “Furthermore, NS5 of JEV and YFV down-regulate STAT2 phosphorylation (87, 97). Interestingly, STAT2-mediated inhibition of YFV is mediated via a unique mechanism dependent on IFN type I stimulation (97). IFN-I first induces STAT1 phosphorylation, and then the E3 ligase TRIM23 promotes NS5 ubiquitination, which allows NS5 to sequester STAT2 only in IFN-267 I-stimulated cells (97).” This did not come across clearly. Is the highlighted part meant to say “STAT2-mediated inhibition by YFV NS5”? Does YFV NS5 antagonise STAT2 by downregulating its phosphorylation as well as sequestering it? How does STAT2 mediate its own phosphorylation and sequestration?
Line 268. “STAT2-mediated sequestration by NS5 of DENV”. I took this to mean that NS5 sequesters DENV and it is mediated by STAT2. Is that correct?
Line 271-273. This sentence could be made clearer with “introduction of human STAT2 gene into the mouse STAT2 locus”.
Line 277-280. “Interestingly, two DENV-1 human isolates with V116A or V116M sequence differences in their NS4B proteins (relative to wild-type DENV-1 NS4B) showed lower levels of IFN-β induction relative to wild-type DENV-1, which may explain the ability of NS4B to inhibit IFN response (100).” It should be mentioned that the two isolates were fitter and had higher virus growth than WT. Instead of “may explain”, do the authors mean “demonstrates”?
Line 305. The acronym LGP2 should be defined.
Line 361-363. It is not clear why one would expect to see changes in expression of RNAi pathway components, assuming ZIKV sfRNA functions by competitively binding with key components of the pathway like Dicer-2 or AGO2. Further, what references 123 and 124 show is that ZIKV sfRNA is needed for ZIKV replication but not through interactions with the RNAi pathway. Although it would interesting point to make that sfRNA of ZIKV supports virus replication differently than sfRNA of DENV-1 or WNV(KUNV), the way this paragraph is currently written does not accurately reflect the source articles.
Line 364-367. This paragraph seems out of place given that the authors have clarified they are focusing only on the exo-siRNA pathway in this review, especially since there is no antagonistic activity to report. The authors could remove this paragraph or expand their scope to include miRNA and piRNA RNAi pathways.
I would have liked to see more of the authors’ perspectives and interpretations of the information they presented. This would increase the value of the manuscript’s contribution to the field and make it stand out from similar reviews. For example, in the conclusion, the authors outlined two open questions that should be addressed between lines 379-385. This was very interesting and more would have been great.
Related to the comment above, the authors have done a good job in highlighting the importance of better understanding antagonistic interactions between flaviviruses and human or mosquito innate immunity. However, the manuscript could be strengthened by providing some concrete implications or research milestones that would pave the way to new antiviral therapies, vaccines, or vector control strategies.
Just a suggestion for an even more comprehensive review: Would the authors consider discussing apoptosis and the signaling pathway that leads to it? Interactions by flaviviral proteins to modulate apoptosis has been reported (examples below) and this would be the principal method of virus clearance in mosquitoes.
- Reference 124
- https://journals.asm.org/doi/full/10.1128/JVI.79.13.8388-8399.2005
- https://www.mdpi.com/1999-4915/9/9/243
- https://www.frontiersin.org/articles/10.3389/fmicb.2021.654494/full
Reviewer 2 Report
The manuscript entitled, Innate Immune Antagonism of Mosquito-Borne Flaviviruses in Humans and Mosquitoes, provides an excellent overview for the field. The article is very topical, well and clearly written, and of great interest for the field of innate immunity of flaviviruses. The differences and similarities between the human and insect innate sensing of flavivirus PAMPs is also well described and cited. One small suggestion would be to improve the discussion in the lines 293 - 310 by including the recent findings of Caetano Reis & Sousa lab (doi: 10.1126/science.abg2264) with the identification of an isoform of Dicer in mammalian stem cells. This review article is very much welcomed and should be accepted for publication.
Reviewer 3 Report
This is a timely and through review of the topic of viral suppression of innate immunity in vertebrates and insects with an appropriate focus on flaviviral pathogens. It will be a useful addition to the literature.
The review is well organized, coherent and concise.
There was only one obvious omission – recent findings by Colmant A, Fros J. et al., (Viruses, 2021) also suggest that the vertebrate Zinc-finger antiviral protein (ZAP) can target the flavivirus genome for degradation, but that vertebrate infecting flaviviruses (but not insect-specific flaviviruses) avoid this response through reduction in frequency of CpG dinucleotides (the binding target of ZAP) their genome. This aspect should also be cited and briefly discussed.
Minor issue:
There are repeated formatting errors associated with some reference citations throughout the MS;
